

# Physiology of pregnancy and oral local anesthesia considerations

Xueer Zhou[1,2,*], Yunyu Zhong[1,2,*], Zijian Pan[1,2], Jiankang Zhang[1,2,3] and Jian Pan[1,2,3]

[1] State Key Laboratory of Oral Disease, West China Hospital of Stomatology, Sichuan University, Chengdu, Sichuan, China
[2] National Clinical Research Center for Oral Diseases and Department of Oral and Maxillofacial Surgery, West China Hospital of Stomatology, Sichuan University, Chengdu, Sichuan, China
[3] Chengdu Advanced Medical Science Center, West China Hospital of Stomatology, Sichuan University, Chengdu, Sichuan, China
* These authors contributed equally to this work.

Corresponding author
Jian Pan, jianpancn@scu.edu.cn

## ABSTRACT

**Background:** Safe and effective local anesthesia is a prerequisite for emergency oral surgeries and most dental treatments. Pregnancy is characterized by complex physiological changes, and increased sensitivity to pain. Pregnant women are particularly vulnerable to oral diseases, such as caries, gingivitis, pyogenic granuloma and third molar pericoronitis. Maternally administered drugs can affect the fetus through the placenta. Therefore, many physicians and patients are reluctant to provide or accept necessary local anesthesia, which leads to delays in the condition and adverse consequences. This review is intended to comprehensively discuss the instructions for local anesthesia in the oral treatment of pregnant patients.
**Methodology:** An in-depth search on Medline, Embase, and the Cochrane Library was performed to review articles concerned with maternal and fetal physiology, local anesthetic pharmacology, and their applications for oral treatment.
**Results:** Standard oral local anesthesia is safe throughout the pregnancy. At present, 2% lidocaine with 1:200,000 epinephrine is considered to be the anesthetic agent that best balances safety and efficacy for pregnant women. Maternal and fetal considerations must be taken into account to accommodate the physiological and pharmacological changes in the gestation period. Semi-supine position, blood pressure monitoring, and reassurance are suggested for high-risk mothers to reduce the risk of transient changes in blood pressure, hypoxemia, and hypoglycemia. For patients with underlying diseases, such as eclampsia, hypertension, hypotension, and gestational diabetes, the physicians should use epinephrine cautiously and control the dose of anesthetic. New local anesthesia formulations and equipment, which contribute to minimizing injection pain and relieving the anxiety, have and are being developed but remain understudied.
**Conclusions:** Understanding the physiological and pharmacological changes during pregnancy is essential to ensure the safety and efficiency of local anesthesia. Optimal outcomes for the mother and fetus hinge on a robust understanding of the physiologic alterations and the appropriate selection of anesthetic drugs and approaches.

# INTRODUCTION

Pregnancy is a particular period during which maintaining oral health is essential. Due to the existence of primary oral diseases and changes in the internal environment, the incidence of oral diseases increases during pregnancy (*Cho et al., 2020*; *Choi et al., 2021*; *Mark, 2021*). Although dental treatment during pregnancy has proven to be safe, due to lack of authoritative clinical consensus and guidelines (*Lee & Shin, 2017*; *Manautou & Mayberry, 2023*), many dentists are hesitant to provide dental therapy to pregnant women or even advise against any treatment during pregnancy (*Lee et al., 2010*). Furthermore, many pregnant women are afraid to seek dental treatment due to dental anxiety and neglect of the importance of oral health. Dental drills and anesthesia injections are the most likely factors for dental anxiety (*Al Khamis et al., 2016*; *George et al., 2018*; *Liu et al., 2019*; *Rocha et al., 2018*).

The mother and the fetus are connected *via* the placenta, and oral health care is particularly important for both. Current clinical practice guidelines clearly state that it's not only safe, but also necessary to provide dental treatment during pregnancy (educational, preventive and restorative actions) (*AAPD, 2023*; *ADA, 2021*; *CDA, 2010*; *Health MDoP, 2016*; *Mark, 2021*; *OHRC, 2023*; *WHO, 2017*). Any improvement in women's oral health and health education can impact children's health (*Figuero, Han & Furuichi, 2020*; *Fomete et al., 2021*; *Raju & Berens, 2021*). Leaving severe inflammation and pain untreated during pregnancy can result in a more harmful emergency that may lead to general anesthesia surgery, hospitalization, and premature delivery (*Bobetsis et al., 2020*; *Cho et al., 2020*; *Favero et al., 2021*; *Fomete et al., 2021*; *Iqbal et al., 2022*; *Kapila, 2021*).

Local anesthesia is a prerequisite for most oral clinical treatments, which can relieve pain in dental procedures such as tooth extraction, fossa preparation, root canal treatment, abscess drainage, and minor oral surgery. Common crises in the oral clinic include syncope, hyperventilation, local anesthetic overdose, hypertension, hypoglycemia, and epileptic seizures, of which more than 50% are associated with local anesthesia injection (*Smereka et al., 2019*). Many local anesthetics and anesthesia methods are commonly used in the clinic, including 4% articaine and 3% mepivacaine infiltration anesthesia, 2% lidocaine, 0.5% bupivacaine, and 2% mepivacaine block anesthesia (*Spivakovsky, 2019*; *St George et al., 2018*; *Wang et al., 2021*). However, pregnancy causes many physiological changes, and maternally administered drugs can affect the fetus through the placenta. Therefore, the selection of anesthetics and anesthesia approaches should be deliberated for the perioperative and intraoperative management of pregnant patients. Herein, this article will comprehensively discuss the instructions for local anesthesia in the oral treatment of pregnant patients and clarify doubts of health professionals, in combination with the physiological changes during pregnancy, local anesthetics pharmacology, advanced materials and equipment, and the process of oral therapy.
## SURVEY METHODOLOGY

An in-depth search on Medline, Embase, and the Cochrane Library was performed to review articles concerned with maternal and fetal physiology, local anesthetic pharmacology, and their applications for oral treatment. A broad range of keywords and phrases were searched, including "pregnancy", "maternal", "fetal", "physiology", "dental" and "oral", "anesthesia", "topical anesthesia", "local anesthetic", "epinephrine", "lidocaine", "efficacy", "pediatric", "pain", "infection", "buffer", "warm", and "equipment", performed by crossing these descriptors using the Boolean operators "OR" and "AND". In the process of summarizing the recommendations for the use of local anesthetics during pregnancy, we further reviewed relevant guidelines from various institutions. Gray literature (monographs, dissertations, theses, books, chapters, studies published in event proceedings) and studies without full text information were excluded. Additional references for possible inclusion were obtained through manual searching. A selection of titles was made, from which the abstracts were read and those that met the topic of the article were selected. A preliminary review of 546 studies was accomplished through electronic database searching, the majority of which were excluded after reviewing the title and abstract. The remaining 237 articles proceeded to the final full-text review and resulted in the inclusion of 182 studies.

### Physiology of pregnancy and anesthesia considerations

Pregnancy is characterized by physiological alterations necessary for fetal growth, some of which may affect local anesthesia efficiency (Fig. 1). Specifically, systemic changes with respect to the cardiovascular, hematologic, respiratory, and nervous systems require careful monitoring. Gastrointestinal, renal, and endocrine alterations may interfere with the determination of anesthesia dose.

### Cardiovascular and hematological alterations

The cardiac output in pregnant patients can increase by 50% in early pregnancy, attributed to peripheral vasodilation, decreased systemic vascular resistance, and a 20% to 30% increase in heart rate (*Aleksenko & Quaye, 2020*; *Yanamandra & Chandraharan, 2012*). Pregnancy is associated with increased sympathetic activity, which may contribute to arrhythmias (*Tamirisa et al., 2022*). As the heart is pushed upward and rotated forward, echocardiography often shows ventricular enlargement and myocardial hypertrophy (*Aleksenko & Quaye, 2020*). Pre-existing cardiovascular conditions such as pre-eclampsia or eclampsia may also progress dramatically during pregnancy.

During the second trimester, pregnant women are at higher risk of hypertension, clinically evident in 8% of pregnant patients (*Barr et al., 2022*), characterized by poorer oral health and a higher incidence of complications with local anesthesia (*Mata et al., 2021*). During dental treatment, fear and pain are the most common causes of high blood pressure. Although treatment of transient hypertension is not recommended in generally healthy patients, severe fluctuations in blood pressure should be avoided in pregnant patients, refraining from hypoxia caused by decreased cerebral and uterine perfusion (*Papademetriou et al., 2021*). If the pregnant woman's intraoperative blood pressure

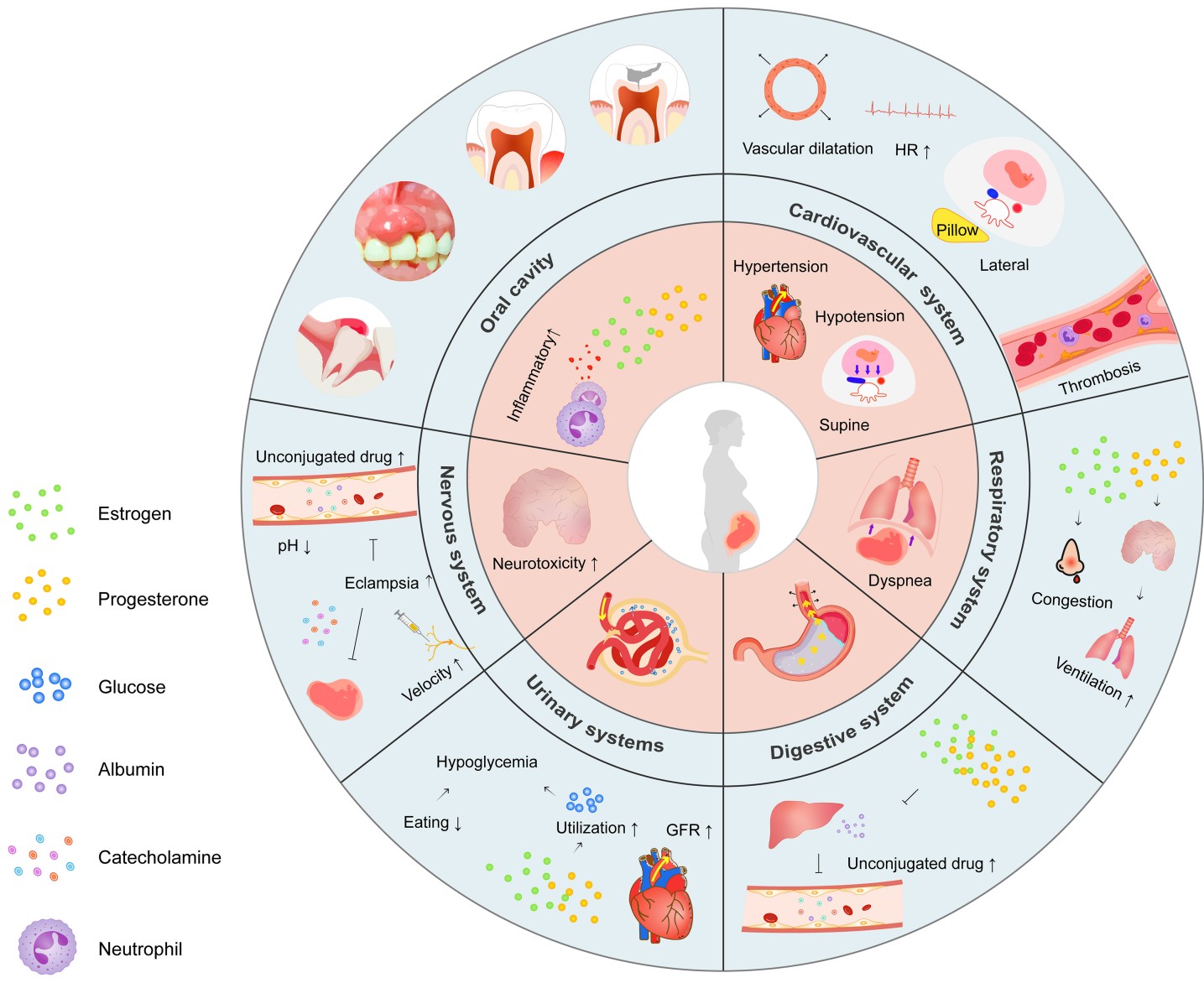

**Figure 1 Schematic illustration of physiological changes during pregnancy.** Overview highlighting the major physiological changes in each system (inner ring) and the mechanisms behind them (outer ring).

suddenly increases, the physicians and dentists should stop the operation immediately, give oxygen in time and closely observe the patient's systemic condition, eliminate the patient's tension and anxiety, and consider whether to continue the treatment according to the patient's recovery (*Southerland et al., 2016*).

Hypotension usually occurs during the second and third trimesters. Hypotension will lead to reduced blood flow to the uterus and fetal perfusion and even lead to fetal hypoxia, so it is necessary to monitor and maintain maternal blood pressure and fluid volume carefully. When the patient is in the supine position, the fetus significantly compresses the inferior vena cava and aorta. As a result, the venous return of cardiac blood volume is impaired, which results in a 14% reduction in stroke output. Blood pressure and cardiac

output are also decreased, resulting in oxygen deprivation to the brain and uterus, which may lead to supine hypotension syndrome. Supine hypotension syndrome occurs in approximately 10% of pregnant patients and is usually characterized by hypotension, syncope, and bradycardia (*Massoth et al., 2022*). The patient should be placed in a semi-supine position or told to lean 5° to 15° to the left (right hip elevation 10 to 12 cm) with a pad on the right lower back to move the uterus toward the aorta, which is less prone to collapse. In addition, sympathetic nerve blocks by local anesthesia may also lead to hypotension. Antihypertensive drugs such as calcium channel blockers, angiotensin II receptor antagonists, and alpha receptor blockers can aggravate the blood pressure reduction caused by local dental anesthesia (*Ouchi & Jinnouchi, 2021*).

During pregnancy, there is an increasing number of white blood cells, red blood cells, and blood clotting factors except for XI and XII, which leads to a fivefold increased likelihood of thromboembolism such as deep vein thrombosis and pulmonary embolism (*Alsheef et al., 2020*; *Guimicheva, Czuprynska & Arya, 2015*). Compression of the inferior vena cava in a supine position also contributes to venous stasis and thrombosis and increases the risk of deep vein thrombosis.

## Respiratory alterations

Progesterone can directly stimulate ventilation by increasing the sensitivity of the respiratory center to carbon dioxide, making gravid women require greater tidal volume to remove carbon dioxide (*Costantine, 2014*; *Kohlhepp et al., 2018*). Hyperventilation begins in the first trimester and may increase to 42% in the third trimester. Moreover, it has been reported that increased serum estrogen concentrations contribute to nasal capillary bleeding, which leads to nasal congestion and breathing difficulties (*Costantine, 2014*). Up to 50% of pregnant women complained of breathing difficulties at 19 weeks of gestation and up to 75% at 31 weeks (*Lee et al., 2017*; *Tan & Tan, 2013*).

In addition to the increased oxygen demands, there are concomitant decreases in oxygen reserves. To compensate for the space taken up by the fetus, the diaphragm moves up, and the residual lung volume drops by up to 20%, especially in the supine position. The mother and fetus are prone to hypoxia due to a decrease in oxygen reserves. Research has demonstrated that moderate hypoxemia occurs in 25% of pregnant women in the supine position. Even short periods of apnea can rapidly lead to maternal hypoxemia and hypercapnea (*Zhao, Wong & Stevenson, 2021*). Therefore, it is necessary to adjust the ventilation pattern and position to avoid hypoxemia.

## Gastrointestinal alterations

The main gastrointestinal changes during pregnancy are related to nausea, vomiting, and heartburn. Increased progesterone levels during pregnancy have been linked to decreased esophageal tone, delayed gastric emptying, and decreased bowel motility, which may lead to gastroesophageal reflux during pregnancy. A supine position can lead to decreased arterial partial oxygen pressure, increasing the risk of gastroesophageal reflux dyspepsia, which can cause aspiration of gastric contents (*Costantine, 2014*). Therefore, if the patient
feels nauseous during the treatment, the operation should be stopped immediately, and the dental chair should be upright.

## Liver, renal and endocrine alterations

The production of estrogen and progesterone increases during pregnancy and reaches a maximum during the third trimester. These hormonal changes can affect liver function. The decreased synthesis capacity of the liver results in decreased levels of albumin and alpha-1 acid glycoprotein, leading to an increase in the proportion of uncombined drugs (*Lim, Mouyis & MacKillop, 2021*). The medications should have their doses reduced accordingly to the increased distribution and enhanced drug action (*Avram, 2020*).

Another typical change in pregnant women is an increased glomerular filtration rate (GFR) due to increased cardiac output (*Bonnet, 2016*; *Habak & Griggs, 2022*). As a result of the increased filtration capacity, the clearance of albumin, glucose, creatinine, and urea increased, as did the clearance of drugs. Drugs such as lidocaine, cleared by the kidneys, should have their doses adjusted to the clearance rates (*Calimag-Loyola & Lerma, 2019*; *Uppal et al., 2022*).

In addition, about 45% of pregnant women develop gestational diabetes during pregnancy (*Chiou et al., 2022*; *Schwartz, Nachum & Green, 2015*). Epinephrine injected during local oral anesthesia can directly increase circulating glucose levels, especially in patients with poor glycemic control (*Byakodi, Gurjar & Soni, 2017*). Therefore, detailed preoperative inquiry and adding epinephrine with caution are necessary.

The incidence of hypoglycemic syncope also increases during pregnancy. Glucose excretion increases in pregnant women due to increased GFR during pregnancy. In addition, estrogen and progesterone can increase the utilization of glucose. The pregnant woman's increased metabolism and the fetus's glucose uptake result in increased glucose requirements. Some pregnant women are also stressed when they feel pain or see dental instruments, which increases the secretion of gastric acid and pepsin in the digestive system and aggravates gastric peristalsis. Under the combined effect of the above factors, pregnant women have lower blood glucose and are susceptible to hypoglycemic syncope.

## Neuropsychological alterations

Adequate and effective psychological counseling is essential for pregnant women before the clinical operation to reduce their fear of treatment. The sensitivity of nerve fibers to local anesthetics during pregnancy is higher than that during non-pregnancy, so the onset time of conduction block is earlier. In addition to the improved anesthesia effect, mental tension and anxiety will be aggravated during pregnancy (*Jarvis et al., 2012*). Increased fear and sensitivity to pain can lead to increased heart rate and blood pressure, which can affect the stability of the fetus and even cause epileptic seizures and abortion (*Benson & Pack, 2020*; *Whelehan & Delanty, 2019*). Seizures are accompanied by increased catecholamine production, followed by decreased blood flow to the placenta. Tissue acidosis and hypoxia are common findings in systemic seizures, which decrease the plasma protein binding rate

of local anesthetic. Once local anesthetics enter the bloodstream, toxic effects on the central nervous system further increase.

## Oral health alterations

Recent studies have shown an association between caries and pregnancy (*Hu et al., 2022*). Specifically, the need for carbohydrates during pregnancy often leads to hunger and increased eating times. Pregnant women inevitably experience vomiting and aggravated pharyngeal reflex, which brings more food residue. The stomach acid contained in this vomit also causes dental erosion. Acidic foods favored by pregnant women can also lower the pH of saliva. Changes in the composition of saliva can temporarily make teeth more prone to erosion and decay (*Doucède et al., 2019*). Moreover, increased estrogen levels in saliva lead to an increased desquamation rate of the oral mucosa, which increases bacterial proliferation and promotes tooth decay (*Yousefi, Parvaie & Riahi, 2020*). Pregnancy also weakens the immune system and affects and accelerates the progression of caries. Caries during pregnancy should be treated in time to avoid abortion and premature delivery caused by severe toothache.

Approximately 60% to 75% of pregnant women have gingivitis, especially in the third trimester of pregnancy (*Vigarios & Maret, 2020*). Due to the increase in estrogen and progesterone levels, insulin sensitivity decreases, and sugar metabolism changes during pregnancy. As a result of the increased blood sugar and volume, the minute vessel expands, elasticity decreases, and blood capillary permeability increases. During pregnancy, the inflammatory response is highly activated, neutrophil function is suppressed, and the expression of inflammatory markers is increased (*Choi et al., 2021*; *Mahapatra et al., 2021*; *Tettamanti, 2017*). Along with the increase in inflammatory cells and exudate, the gums are often congested and swollen with hypertrophic inflammation during pregnancy (*González-Jaranay et al., 2017*; *Vigarios & Maret, 2020*). Supposing there is a large amount of plaque and calculus accumulation in this period, gingival inflammation will be heightened and finally form periodontal disease. Periodontal treatment is encouraged to alleviate suffering and reduce the risk of premature birth and other adverse reactions during pregnancy (*Iheozor-Ejiofor et al., 2017*; *Iqbal et al., 2022*).

Another pathological gum state that occurs during pregnancy is pyogenic granuloma, with an incidence of about 3% to 10%. Lesions are similar to benign tumors caused by the growth of connective tissue in the gums and most often occur in areas of inflammation or chronic trauma in the gums (*Sarwal & Lapumnuaypol, 2022*; *Silva de Araujo Figueiredo et al., 2017*). Clinical cleaning and curettage are often used to remove local irritation in the affected area. Prompt surgical removal is required for large gingival tumors with persistent pain or infection (*Guastella et al., 2017*).

Third molar pericoronitis can cause not only affliction but also fever and fascial space infection. Patients have difficulty eating, chewing, and swallowing and restricted mouth opening, which can cause infection of adjacent tissues and organs or interstitial space in severe cases. The treatment choices for mild symptoms are local irrigation to control symptoms or removal under close supervision. In the case of interstitial infection, abscess incision and drainage are recommended under anesthesia.

Emergency oral treatment during pregnancy is often accompanied by local acute inflammation, which causes difficulty in anesthesia. Research has demonstrated that the success rate of lidocaine block anesthesia in patients with pulpitis is only 67%, much lower than that in normal teeth (*Visconti, Tortamano & Buscariolo, 2016*). Inflammation and bacteria can lead to increased expression of neuropeptides such as Substance P and calcitonin gene-related peptides, as well as the release of inflammatory mediators such as prostaglandin E2, prostaglandin F2α, and interleukin 1 and 6, leading to the excitement of pain receptor subtypes. Moreover, expression of the sodium channel increases in inflamed dental pulp, which results in the alteration of resting membrane potential, and the decreased excitability threshold of the teeth (*Wang et al., 2021*). Clinical manifestations include changes in neuronal plasticity, abnormal pain, and peripheral and central hyperalgesia (*Drum et al., 2017*). The acidic tissue environment of inflammation also makes it difficult for drugs to be converted into deionized forms of active free radicals, making it difficult for lidocaine molecules to cross nerve membranes, resulting in poor anesthesia effects (*Yilmaz, Tunga & Ozyurek, 2018*).

## Fetal considerations

Most drugs given to pregnant women may affect the fetus after they are transferred across the placenta and enter the systemic circulation of the fetus. The effects of the transferred drug on the fetus depend on the type of drug and the general fetal conditions (*Bouazza et al., 2019*; *Kolding, Eken & Uldbjerg, 2020*; *van Hove et al., 2022*). Therefore, it is essential to understand the pharmacology of local anesthetic, the most commonly used drugs in dental treatment, for safe and effective dental treatment during pregnancy (*Ouanounou & Haas, 2016*).

The severity of the effects of local anesthetic on the fetus depends on the dose, whether a vasoconstrictor is used, the metabolic rate of local anesthetic, the degree of protein binding, and pKa (acid dissociation constant). The degree of binding of local anesthetic to maternal protein has the most significant effect. Only unconjugated anesthetic molecules can be transferred across the placenta to the fetus (*Pemathilaka, Reynolds & Hashemi, 2019*). For the fetus, the potential risks associated with anesthesia are mainly fetal hypoxia, miscarriage or premature delivery, and to a lesser extent teratogenicity (*Tolcher, Fisher & Clark, 2018*; *Vasco Ramirez & Valencia, 2020*).

## Teratogenicity

"No currently utilized anesthetic drugs have been found to have any teratogenic effects in humans when utilizing standard concentrations at any gestational age," claims the *American College of Obstetricians and Gynecologists (2017)*. Multiple clinical studies have demonstrated that oral local anesthesia and oral treatment during pregnancy does not increase the risk of fetal malformation within the maximum safe dose (*Hagai et al., 2015*; *Michalowicz et al., 2008*; *Moore, 2016*). Herein, dentists who are cautious and reluctant to use local anesthetics on pregnant women should change their views and believe that women can receive necessary dental treatment during pregnancy as long as the anesthetic is chosen correctly and the dosage is controlled.

## Hypoxia and asphyxia

The most severe adverse fetal event of anesthesia during pregnancy is intrauterine asphyxia. Due to the lack of automatic regulation of placental perfusion and the complete dependence of fetal oxygenation on maternal oxygenation, any hemodynamic changes in the mother will directly affect uteroplacental perfusion. Prolonged or severe maternal hypotension, hypoxemia, and hypercapnia can result in decreased uterine placental blood flow and fetal ischemia (*Lato et al., 2018*). Injecting small amounts of local anesthetics into the head and neck region may also result in unfavorable effects that are comparable to systemic toxicity caused by inadvertent intravascular injections of higher dosages. Cardiovascular stimulation or depression, disorientation, seizures, and respiratory depression have been reported. These reactions could be the result of intra-arterial local anesthetic injections with retrograde cerebral circulatory flow (*Johnson, Boscoe & Cabrera-Muffly, 2020*). Epinephrine can reduce blood flow in the uterus and uterine contractile force proportionally to the dose when given intravenously (*Hood, Dewan & James, 1986*), further inducing fetal hypoxia (*Badran et al., 2019*; *Ritchie et al., 2017*). Although there have been no cases of oral local anesthetic causing fetal hypoxia, oral local anesthetic may do so in expectant mothers (*Tomlin, 1974*). Whilst under block anesthetic, patients should have their breathing and circulation closely monitored. Dose guidelines shouldn't be surpassed. In anoxic fetuses, the binding of the local anesthetic to the protein is reduced compared to healthy fetuses, and lidocaine is retained due to tissue acidosis (*Ralston & Shnider, 1978*). Thus, fetal sensitivity to neurotoxicity and cardiovascular toxicity of local anesthetic increases. Therefore, extra attention should be given to maintaining maternal blood pressure and oxygenation. Aspiration should be done before to injecting the local anesthetic solution in order to prevent intravascular injection. The needle must be adjusted until aspiration produces no return of blood. For fetuses at high risk of asphyxia or in poor general condition, local anesthetic and epinephrine must be used with caution (*Bonnet, 2016*).

## Miscarriage and premature birth

Sporadic spontaneous abortion occurs most often in the first trimester of pregnancy. Premature delivery is considered under 37 weeks (*Purisch & Gyamfi-Bannerman, 2017*). Although local anesthesia has no observed adverse impact on healthy fetuses, the risk is higher for pregnant women with medical conditions such as heart disease and pre-eclampsia. Pre-eclampsia is a pregnancy complication characterized by high blood pressure, proteinuria, and edema in 3–7% of pregnant women (*Rana et al., 2019*). On the one hand, there is a maternal syndrome characterized by endothelial cell activation, blood pressure disturbance, proteinuria, and edema; on the other hand, there is reduced intrauterine growth of the fetus (*Phipps et al., 2016*; *Rana et al., 2019*). Pre-eclampsia in pregnant women has reduced protein binding to local anesthetic. When placental perfusion is suddenly reduced, many free forms of local anesthetic can be transferred to the fetus, resulting in premature delivery.

Therefore, the type and dosage of local anesthetic must be carefully considered for pregnant women with underlying health problems. The management of anesthesia in

pregnant women should focus on the hemodynamic stability of the mother to avoid hypoxemia and alkalosis or acidosis (*Ralston & Shnider, 1978*). Abdominal pain during the procedure may indicate premature labor or miscarriage. If this occurs, dentists should stop the operation immediately and call the obstetrician for help (*Vasco Ramirez & Valencia, 2020*).

## PREOPERATIVE PROPHYLAXIS

### The timing of the operation

Currently, it is not recommended to postpone dental treatment during pregnancy (educational, preventive, and restorative actions), while significant traumatic periodontal surgery should be avoided during pregnancy. The best time for dental treatment is between 14 and 20 weeks of pregnancy (*OHRC, 2023*). The fundamental reason for this approach is that pregnant women will experience considerable psychological and physiological changes that they have not fully adapted to during early pregnancy. In addition, many pregnant women experience severe morning sickness during the first trimester, which complicates clinical therapy (*CDA, 2010*). In the third trimester of pregnancy, the enlarged uterus makes it difficult to lie in the chair for a long time, which increases the difficulty of complex oral treatment. The supine position is more prone to inferior aortic vena cava compression and lead to postural hypotension, which needs to be alleviated by placing the pregnant woman in a semi-decumbent position and changing the position frequently. The recommendation for the second trimester is based on psychological and physical comfort and does not imply that oral therapy should be prohibited in the first and third trimesters. Prompt treatment of oral diseases and infections is beneficial to women at all stages of pregnancy. Pregnancy should not be a reason to delay treatment for oral emergencies at any point in pregnancy, as acute dental pain or infection can have serious adverse consequences for the pregnant woman and fetus (*Kapila, 2021*).

### Preoperative assessment of physical and psychological status

The combination of fear of LA injection and oral therapy can cause severe psychological discomfort and sometimes potentially life-threatening conditions, such as vascular suppression syncope, hyperventilation, clonic convulsions, bronchospasm, angina pectoris, *etc.* (*Armfield & Heaton, 2013*). Physicians and dentists should inquire about medical history and pregnancy-related causes in detail, paying particular attention to any conditions that raise the patient's blood pressure.

### Local anesthesia agents

To determine the risks associated with drugs used during pregnancy, the United States Food and Drug Administration (*FDA, 1979*) classified drugs according to their risk level to the fetus (Table 1). Categories A and B are considered safe with no adverse effects on the fetus. Drugs in category C have been shown to have teratogenic risk in the animal fetus, but there is no adequate evidence in humans (*Lee & Shin, 2017*). Lidocaine and prilocaine are rated B by the FDA and are considered the safest local anesthetic for pregnant women. Because the concentration of prilocaine (4%) is higher than that of lidocaine (2%),

**Table 1 Summary of local anesthetics used in pregnant dental patients (*FDA, 1979*).**

| Agent | FDA category | Using during pregnancy |
|---|---|---|
| Articaine | C | Use with caution |
| Bupivacaine | C | Use with caution |
| Benzocaine | C | Use with caution |
| Mepivacaine | C | Use with caution |
| Lidocaine | B | Safe |
| Prilocaine | B | Safe |

resulting in more drugs being administered per injection, lidocaine is the preferred choice in the clinic.

Another commonly used local anesthetic, articaine, is in category C (*FDA, 2023*). The combination of articaine hydrochloride and epinephrine has been proven to increase fetal mortality and skeletal variation in rabbits when administered subcutaneously at four times the maximum recommended dose (FDA). The FDA recommends that articaine hydrochloride and epinephrine should only be used during pregnancy if the potential benefits outweigh the potential risks to the fetus.

## Pharmacological characteristics of lidocaine

After injection, lidocaine is buffered by body fluids and dissociated into an uncharged lipid-soluble base and a relatively lipid-insoluble cation in equilibrium. The base form penetrates the nerve membrane into the nerve axon. The amount of base type is closely related to the pKa of local anesthetic and the pH value of tissues, according to the Henderson–Hasselbalch equation, $pH = pKa + \log(base/acid)$. After entering the nerve membrane, the free base form is converted back to the cation form, binds to the sodium channel, reduces the membrane's permeability to sodium ions, and induces a non-depolarized nerve block (*Malamed, 2014*).

The liver metabolizes 90% of lidocaine, and 10% is excreted in its original form (*FDA, 2021*). Because lidocaine hydrochloride is metabolized quickly and can dilate blood vessels, most drugs are absorbed into the body quickly. This condition not only shortens the duration of anesthesia but also increases the risk of intoxication. Therefore, adrenaline is often added to lidocaine solutions to extend the duration and reduce the risk of poisoning (*Ogawa et al., 2021*).

## The epinephrine controversy

FDA classifies epinephrine as Pregnancy Category C because it is teratogenic in rabbits, mice, and hamsters dosed during organogenesis. Epinephrine should be used during pregnancy only if the potential benefit justifies the potential risk to the fetus (fetal anoxia, spontaneous abortion, or both) (*FDA, 2022*). Although epinephrine is injected outside the blood vessel, it has been shown to quickly enter the circulatory system after oral injection,

with a peak occurring about 8 min after injection (*Takahashi et al., 2005*). Epinephrine is influential in various systemic diseases such as hyperthyroidism, glaucoma, and diabetes (*FDA, 2022*). Epinephrine, on the other hand, can significantly constrict blood vessels in the uterus and reduce blood flow to the placenta. In pregnant women with peripheral arterial obstructive and hypertensive vascular disease, epinephrine may exhibit a significant vasoconstriction response leading to ischemic injury or necrosis and must be used with caution. The increased risk may be even more significant in patients with severe cardiovascular disease or those taking medications that interact with epinephrine (*FDA, 2022*).

Most current studies considered that using epinephrine for LA during pregnancy is safe as long as the potential risks of intravascular injection are minimized and safe injection doses and methods are guaranteed (*Ilyas et al., 2020*). Clinical studies have proven that low-dose epinephrine used in dentistry does not significantly affect arrhythmia incidence, mean arterial pressure, heart rate, or other indicators, even in patients with cardiovascular disease (*Guimaraes et al., 2021*; *Tschopp et al., 2018*).

The improvement of the anesthesia effect with the addition of epinephrine is significant. One advantage is local vasoconstriction, resulting in a prolonged duration of anesthesia (*Tschopp et al., 2018*), reduced bleeding at the site of administration, and the increased concentration of local anesthetic in pulp tissue (*Fujita & Sunada, 2021*; *Tanaka et al., 2016*). The addition of epinephrine also delayed the absorption of the local anesthetic, resulting in a gradual increase in lidocaine levels in the blood with no peak and a peaceful transfer to the fetus with increased safety (*Lee & Shin, 2017*).

The concentration of epinephrine added to 2% lidocaine is still controversial, and the dosage form of 1:80,000–1:200,000 epinephrine is more commonly used clinically (*Singla et al., 2022*). Higher concentrations, such as 1:50,000, do not provide faster onset or longer duration for local oral anesthesia (*Dagher, Yared & Machtou, 1997*; *Wali et al., 2010*). Although high epinephrine concentrations can help reduce bleeding at the site of local invasion, there is a greater risk of acute epinephrine reactions such as hypertension and tachycardia (*Giovannitti, Rosenberg & Phero, 2013*). According to currently limited studies, it is safer to inject 2% lidocaine with 1:200,000 epinephrine. Although the anesthesia effect is slightly lower than that of the injections of 2% lidocaine with 1:80,000 epinephrine, there is no significant difference (*Aggarwal et al., 2020*; *Karm et al., 2017, 2018*).

To minimize the cardiovascular risks associated with epinephrine, clonidine, a hypertensive drug that can be used in pregnant women, may be an effective and safe alternative to epinephrine for LA during pregnancy. Multiple studies have shown that clonidine improves the success rate of lower alveolar nerve block anesthesia in patients with symptomatic irreversible pulpitis (*MacDonald et al., 2021*; *Shadmehr et al., 2017*), reduces postoperative pain and analgesic use (*Shadmehr et al., 2017, 2021*), and, most importantly, reduces cardiovascular risk compared with epinephrine (*Chowdhury, Singh & Shah, 2012*; *Jimson et al., 2015*; *Patil & Patil, 2012*).

## INTRAOPERATIVE PROPHYLAXIS

### Improved anesthesia techniques

Discomfort and pain from LA injection may rapidly release large amounts of catecholamine, causing severe adverse reactions. Topical anesthesia before injection can eliminate or minimize pain caused by the needle, and undoubtedly add to the safety of LA in pregnant patients. Topical anesthesia and local cooling are commonly used to achieve surface tissue anesthesia in clinics (*Maia et al., 2022*; *Patel et al., 2021*). The commonly used topical anesthetics in the clinic include 8% lidocaine gel, 5% EMLA (2.5% lidocaine and 2.5% prilocaine) cream, and 20% benzocaine, among which only lidocaine and prilocaine are rated as B by the FDA and can be used during pregnancy. Moreover, precooling the injection site utilizing popsicle sticks, frozen cotton swabs, and ice packs is a simple, reliable, low-cost, and practical clinical method to reduce patient suffering (*Amrollahi, Rastghalam & Faghihian, 2021*). The analgesic effect is similar to that of 5% lidocaine gel, while the unpleasant taste of the gel is avoided (*Hindocha et al., 2019*). Vasoconstriction induced by precooling reduces tissue metabolism and the inflow of inflammatory mediators during needle penetration (*Taylor et al., 2019*). In addition, ice massage also stimulates A-δ fibers and activates inhibitory pain pathways, thereby delaying or eliminating pain signal transmission and raising pain thresholds (*Anantharaj et al., 2020*; *Hameed et al., 2018*; *Jayasuriya, Weerapperuma & Amarasinghe, 2017*; *Lakshmanan & Ravindran, 2021*).

Local infiltrating anesthesia is the primary injection method for maxillary anesthesia because of its high success rate and low risk of injection into blood vessels (*Sandilya et al., 2019*; *Wang et al., 2021*). However, the mucosa of the palate is close to the periosteum, and there is an extensive neural network, so infiltration anesthesia in the palatal region is painful and irritating. Multiple studies have shown that although the efficacy of buccal infiltration anesthesia of 2% lidocaine with epinephrine for maxillary molar extraction is still controversial, it can provide similar effects to buccal-palatal infiltration anesthesia in other maxillary dental regions (*Deshpande et al., 2020*; *Phyo et al., 2020*; *Rayati et al., 2021*). The effect of 2% lidocaine infiltration on the mandible is not sufficiently stable. Although lidocaine infiltration anesthesia has been reported to be sufficient for the extraction of mandibular incisors and premolars (*Ege & Demirkol, 2021*; *Jamil, Asmael & Al-Jarsha, 2020*), anesthesia for mandibular molars has a lower success rate or needs to be combined with other injection methods (*Gazal et al., 2020*). Attention should also be paid to the location of the pinhole, and the injection point should be as close to the nerve foramen as possible (*Wang et al., 2021*).

Nerve block anesthesia has incomparable advantages such as less anesthetic dosage, wider anesthesia range, and longer effective anesthesia time, and is especially suitable for a long time and a wide range of surgeries, such as mandibular third molar extraction. The downside is that the failure rate of inferior alveolar nerve block anesthesia (IANB) is up to 20–47% in symptomatic patients with irreversible pulpitis (*Crowley et al., 2018*). If routine IANB fails, patients with irreversible pulpitis often need supplementary injections such as intra-osseous anesthesia (*Yadav, 2015*), periodontal injection (*Gupta*

*et al., 2022*; *Wadia, 2022*), and infiltrative anesthesia (*Gazal et al., 2020*; *Gupta et al., 2021*, *2022*) to achieve complete anesthesia. In addition to conventional nerve block anesthesia, Goe-Gates effectively improved the success of IANB by moving the injection point up to the neck of the condyle (*Aggarwal, Singla & Kabi, 2010*). At the same time, the positive rate of blood recovery can be reduced to approximately 2% (*Sokhov, Rabinovich & Bogaevskaya, 2019*).

## New local anesthesia formulations

Topical anesthesia offers the possibility of painless dental anesthesia, but there are currently no commercially available formulations that eliminate the pain of local anesthesia injections (*Hindocha et al., 2019*). In order to enhance the permeability of lidocaine and improve the effectiveness of surface anesthesia, some materials, such as liposomes (*Franz-Montan et al., 2015*; *Ribeiro et al., 2016*), silica nanoparticles (*Nafisi et al., 2018*; *Sato et al., 2019*), and chitosan (*Qi et al., 2021*), have been gradually developed as lidocaine delivery systems. However, most lidocaine delivery systems focus on the transdermal route. Only a few studies have reported using the lidocaine liposome complex in the oral mucosa.

*Ribeiro et al. (2016)* evaluated the package formulation of the most effective lipid carrier for delivering 2.5% lidocaine and 2.5% prilocaine based on a factored design data analysis program. The prepared eutectic mixture loaded with pectin and liposomes has the advantages of long drug release time, high permeability, and long anesthesia time. Subsequently, yellow collagen was combined with the lipid carrier mentioned above to improve the adhesion of oral mucosa (*Ribeiro et al., 2018*). *Cordeiro Lima Fernandes et al. (2021)* also prepared a kind of prilocaine-lidocaine lipid nanogel, which also had the advantages of small particle size, good dispersion, and adhesion to oral mucosa and could effectively prolong the anesthesia time. A clinical study has also demonstrated that the lidocaine-prilocaine liposome complex can provide adequate palatal mucosal topical anesthesia for maxillary molar extraction without mucosal discomfort, allowing the physician to obtain complete anesthesia for approximately 26 min without injection (*Amorim et al., 2020*).

In addition to the combination with prilocaine, lidocaine oral mucosal patches can enhance surface anesthesia efficacy. *Roh et al. (2016)* developed an oral adhesive bilayer containing lidocaine that enabled 80% of lidocaine to be released within 1 min ($p < 0.05$), effectively permeate the mucosa ($p < 0.05$), and remain attached for at least 3 h. Microneedle patches applied to the oral mucosa before lidocaine injection are also reported to be effective in reducing the pain of injection (*Daly et al., 2021*). 3D printing has been used to create a customizable lidocaine-loaded patch that perfectly attaches to the tooth for more than 1 h while releasing lidocaine from the hydrogel (*Ou et al., 2019*).

## Injection dose

Pregnant patients have a higher risk of suffering from local anesthesia intoxication due to complex physiological changes, making the control of anesthetic dose particularly crucial. Fortunately, clinical studies have shown that using local dental anesthesia during

**Table 2 The maximum recommended doses of lidocaine (*FDA, 2021*; *Fleisher Lee et al., 2014*).**

| Volume of cartridge | Agents | Concentration | | Maximum dosing | | | Maximum number of cartridges | | |
|---|---|---|---|---|---|---|---|---|---|
| | | Lidocaine (mg/cartridge) | Epi (mg/cartridge) | Lidocaine (mg) | Lidocaine (mg/kg) | Epi (mg) | 50 kg patient | 65 kg patient | 80 kg patient |
| 1.8 ml | 2% lidocaine, 1:100,000 epi | 36 | 0.018 | 500 | 7 | 0.2 | 9 | 11 | 11 |
| | 2% lidocaine, 1:80,000 epi | 36 | 0.0225 | 500 | 7 | 0.2 | 8 | 8 | 8 |
| | 2% lidocaine, 1:50,001 epi | 36 | 0.036 | 500 | 7 | 0.2 | 5 | 5 | 5 |
| | 2% lidocaine, plain | 36 | – | 300 | 4.5 | – | 6 | 8 | 8 |
| 5 ml | 2% lidocaine, 1:100,000 epi | 100 | 0.05 | 500 | 7 | 0.2 | 3 | 4 | 4 |
| | 2% lidocaine, 1:80,000 epi | 100 | 0.0625 | 500 | 7 | 0.2 | 3 | 3 | 3 |
| | 2% lidocaine, 1:50,000 epi | 100 | 0.1 | 500 | 7 | 0.2 | 2 | 2 | 2 |
| | 2% lidocaine, plain | 100 | – | 300 | 4.5 | – | 2 | 2 | 3 |

**Note:**
Abbreviations: epi, epinephrine.

pregnancy does not increase the risk of fetal malformation at normal dental doses (*Fisher et al., 2020*; *Hagai et al., 2015*; *Moore, 2016*). Therefore, dentists who are cautious and reluctant to use local anesthesia on pregnant women should revise their views about dental treatment for pregnant women. Women can receive necessary dental treatment during pregnancy if they choose the proper anesthetic and control the dosage correctly.

The American Dental Association Dental Treatment Guidelines state that dentists should use the lowest concentration and volume of anesthetic fluid that provides adequate anesthesia (2009). The maximum dosage of lidocaine may be calculated to be 4.5 mg/kg, and the total dose should be at most 300 mg. The dose of lidocaine in combination with epinephrine should not exceed 7 mg/kg, and the maximum total dose should not exceed 500 mg (*FDA, 2021*). Moreover, the maximum dosage of epinephrine when used with local anesthetic is less than 0.2 mg for healthy patients and less than 0.04 mg for patients with heart disease (*FDA, 2022*; *Fleisher Lee et al., 2014*). The maximal dose of lidocaine for local anesthetic is presented in Table 2 (*FDA, 2021*; *Fleisher Lee et al., 2014*).

# HOW TO MINIMIZE INJECTION PAIN

## Buffering

As described above, the molecular and dissociated lidocaine ratio is determined by the anesthetic's pH and pKa and the tissue's pH. The pH of most lidocaine preparations is reduced to 3–4 by hydrochloric acid because the charged form of the lidocaine molecule is more stable at low pH (*Malamed, 2014*). However, the acidification of lidocaine has increased the sting on injection and the onset of deep anesthesia, and reduced the anesthesia effect of numbing inflamed or infected teeth (*Malamed, 2014*; *Yilmaz, Tunga & Ozyurek, 2018*). Theoretically, the ability of anesthetic molecules to penetrate the nerve

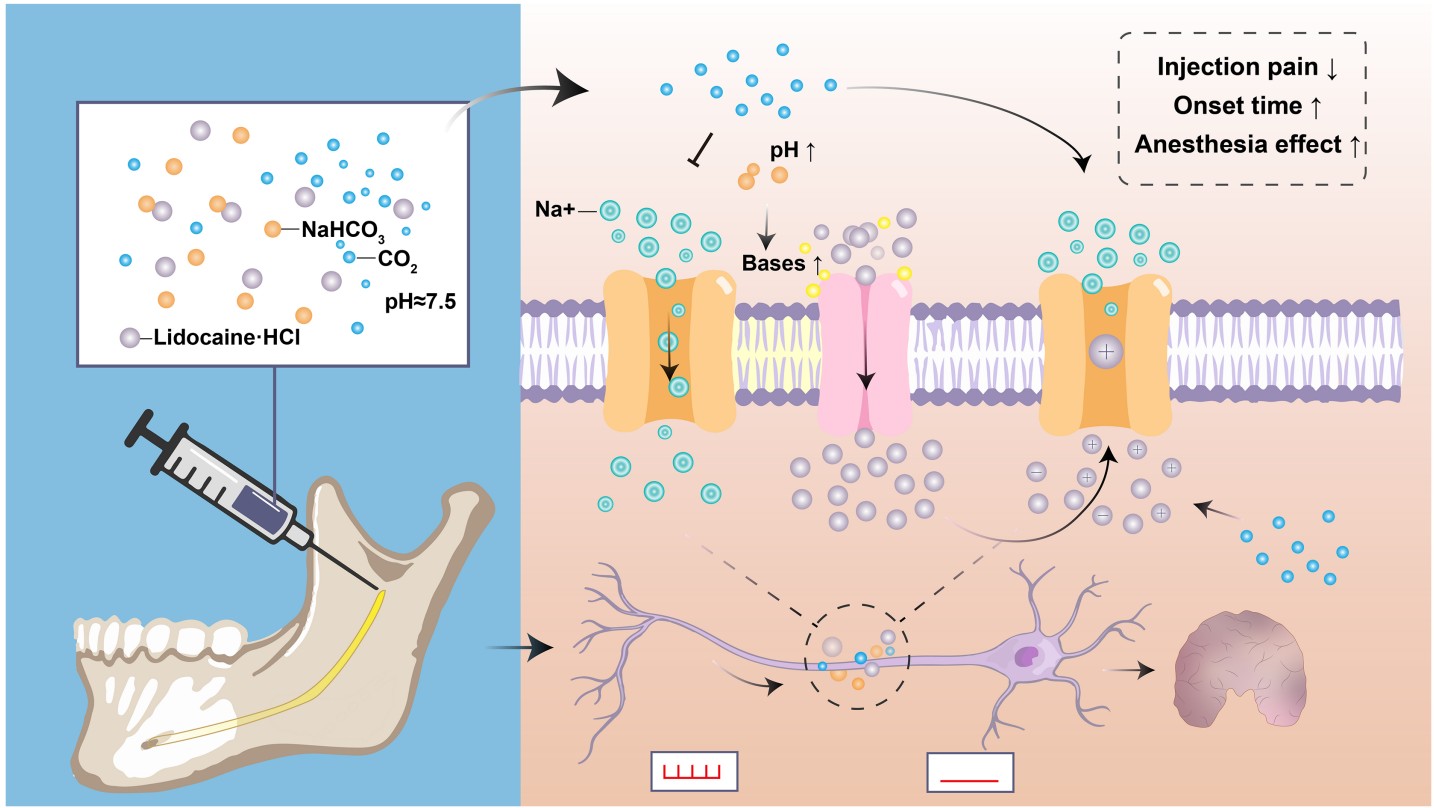

**Figure 2 Diagram depicting the applications and mechanisms of sodium bicarbonate buffering.** Sodium bicarbonate buffering raises anesthesia pH, and the rapid formation of a mixture of charged and uncharged forms leads to more rapid drug diffusion and faster onset of nerve block, and makes injections more comfortable. Carbon dioxide as a byproduct may enhance anesthesia by directly inhibiting nerve conduction and by concentrating local anesthetic through ion retention.

membrane depends on base concentration, which is enhanced when the increase in the number of bases at high pH allows more anesthetic to exist in their non-ionizing form. The active form of free radicals penetrating the nerve membrane into the inner part of the neuron is increased, thus shortening the onset time and reducing the pain of injection (*Amorim et al., 2021*; *Arora, Degala & Dasukil, 2019*). Additionally, when the sodium bicarbonate solution is mixed with lidocaine, the sodium bicarbonate interacts with hydrochloric acid in the anesthetic to produce water and carbon dioxide. Carbon dioxide can enhance local anesthesia by directly inhibiting the nerve axons, increasing the concentration of lidocaine in the nerve stem, and changing the charge of local anesthesia in the nerve (Fig. 2) (*Catchlove, 1972*).

Multiple clinical trials have also supported the theory that sodium bicarbonate reduces the pain of injection and improves the anesthesia effect by buffering lidocaine (*Kattan et al., 2019*; *Kurien, Goswami & Singh, 2018*). Interestingly, a study of 96 patients with periapical infections requiring tooth extraction showed that a single injection of 0.5 ml 8.4% sodium bicarbonate significantly relieved pain after lidocaine anesthesia failed ($p < 0.1$) (*Senthoor, Janani & Ravindran, 2020*). However, there have also been clinical studies showing different results (*Aulestia-Viera, Braga & Borsatti, 2018*; *Guo et al., 2018*).

Since the therapeutic use of sodium bicarbonate as an alkalizing agent for local anesthetic has not been approved, it should be used with caution.

## Warming

The pKa value of local anesthetic will decrease with increasing temperature. Under the condition in constant pH, the decrease of pKa will increase the proportion of free radicals from drugs, and thus increase their passive diffusion in non-neural structures, helping to block the transmission of pain signals (*Ince et al., 2021*; *Lim et al., 1992*). In addition, increased temperature also increases the fluidity of the lipid membrane, allowing lidocaine to penetrate the membrane and reach effective concentrations for faster analgesia. The enhancement of the anesthesia effect may also be related to the transient receptor potential vanilloid (TRPV1) channel in trigeminal nerve tissue, which is activated in near-harmful temperature ranges (over 42 °C). Once activated, the widened membrane gap facilitates the infiltration of cations, thereby increasing the concentration of lidocaine molecules within the nerve cells (*Binshtok et al., 2009*; *Caterina et al., 1997*; *Zhou et al., 2020*).

Clinical results have also demonstrated that the use of warmed lidocaine for inferior alveolar nerve block anesthesia results in a faster onset of anesthesia ($p = 0.004$) and less pain ($p < 0.001$) in patients (*Kurien, Goswami & Singh, 2018*). *Tirupathi & Rajasekhar (2020)* included four studies from 2018 to 2020 to evaluate the subjective and observed pain reactions and found that heating the local anesthesia solution to body temperature (37 °C) before administration reduced discomfort during oral local anesthesia administration.

Moreover, a significant source of injection pain is subcutaneous tissue expansion. Therefore, slow injection has a significant effect in reducing pain (*Garret-Bernardin et al., 2017*). When used for oral treatment under local anesthesia, lidocaine should be injected at less than 1 ml/min (*FDA, 2021*; *Tangen et al., 2016*). In addition, distracting methods such as kneading the patient's cheeks have been proven to relieve pain during anesthesia injections (*Birnie et al., 2018*; *Chen et al., 2020*).

## Applicable syringes

Currently, many innovative dental syringes, such as computer-controlled autoinjectors, are gradually being applied clinically to reduce the discomfort of standard local anesthesia. The vibration technique was first applied to reduce pain. The impression of "pain" from the pressure of the fluid entering the tissue is lessened when the brain activity is concentrated on the vibration (*Nanitsos et al., 2009*). Vibration and touch also stimulate inhibitory interneurons in the spinal cord, which transmit information to second-order neurons in the spinal cord *via* the A-δ and C fibers, and ultimately lead to pain elimination (*Dickenson, 2002*). Computer-aided devices, such as VibraJect, Dental Vibe, Accupal, and Wand, can significantly reduce pain at the injection site, relying on machine vibrations (*Bilsin, Güngörmüş & Güngörmüş, 2020*; *Joshi et al., 2021*; *Midha et al., 2021*; *Salma et al., 2021*).

Intraosseous anesthesia is the injection directly into the cancellous bone near the tooth to be anesthetized. It can achieve deep anesthesia without numbness of the lip and cheek tissue and is mainly used in root canal practice. Intraosseous anesthesia can significantly enhance the effect of pulp anesthesia in patients with irreversible pulpitis and can be used for supplementary anesthesia after the failure of IANB alone (*Dias-Junior et al., 2021*; *Kc, Bhattarai & Subedi, 2022*; *Zanjir et al., 2019*). The continuous improvement of intraosseous anesthesia equipment is receiving increasing attention due to its good anesthesia effect, minimal invasiveness, simple operation, and easy control (*Sovatdy et al., 2018*).

## CONCLUSIONS

Although pregnancy is a particular event characterized by systemic alterations, it is necessary and safe to treat oral diseases that require treatment. Local anesthetics remain the safest and most efficient agents in dentistry to relieve intraoperative and postoperative pain. Local anesthesia is safe for both mother and fetus as long as a safe anesthetic is chosen and detailed guidelines are followed. Emerging injection techniques and anesthetic formulations have proven effective in a more extensive age range. Nevertheless, pain is a multi-factorial problem. It is still necessary to evaluate the effect of the new technology on pregnant women before clinical promotion. Joint efforts should be made to develop an obstetrician-dental document, which is necessary to remove doubt, uncertainty, or unnecessary suffering.

### Funding

This work was supported by the project of the Research and Develop Program, West China Hospital of Stomatology Sichuan University (LCYJ 2019-1), the Science and Technology Project of the Health Planning Committee of Sichuan (21PJ062), and the Natural Science Foundation of Sichuan Province (2022NSFSC1462). The funders had no role in study design, data collection and analysis, decision to publish, or preparation of the manuscript.

### Grant Disclosures

The following grant information was disclosed by the authors:
Research and Develop Program, West China Hospital of Stomatology Sichuan University: LCYJ 2019-1.
Science and Technology Project of the Health Planning Committee of Sichuan: 21PJ062.
Natural Science Foundation of Sichuan Province: 2022NSFSC1462.

### Competing Interests

The authors declare that they have no competing interests.

### Author Contributions

- Xueer Zhou conceived and designed the experiments, performed the experiments, analyzed the data, prepared figures and/or tables, and approved the final draft.

- Yunyu Zhong performed the experiments, analyzed the data, prepared figures and/or tables, and approved the final draft.
- Zijian Pan performed the experiments, analyzed the data, prepared figures and/or tables, and approved the final draft.
- Jiankang Zhang conceived and designed the experiments, authored or reviewed drafts of the article, and approved the final draft.
- Jian Pan conceived and designed the experiments, authored or reviewed drafts of the article, and approved the final draft.

## Data Availability

This is a literature review.

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
