# Peer review of "Physiology of pregnancy and oral local anesthesia considerations"

_PeerJ, doi:10.7717/peerj.15585_

## Round 0.1 · original submission · Major Revisions

Dear Authors, the topic is very interesting and very useful and for this reason, it is necessary to respond to what the reviewers requested.

·

Basic reporting

Thank you for providing this review article on the use of local anesthetics in pregnant women. This review is of broad and cross-disciplinary interest and within the scope of PeerJ. It is written in clear English throughout. The cited references embed the article in the appropriate context. The need to become aware of the benefits and risks of using local anesthetics is presented.

The work is clearly structured. The figures and tables have been labelled sufficiently. Especially figure 1 is very valuable as a graphical summary of the review results.

The difference in meaning between the inner red circle and the outer blue circle is not self-explanatory and requires an additional note.

Experimental design

The survey methodology needs further explanation. Although the searched databases were documented, it is necessary to list the search terms as well in order to be able to reproduce the search results. To get an impression of the representative strength of the review, it would be useful to list the number of articles screened and describe the selection process that led to the 206 referenced papers.

Validity of the findings

The review provides primarily informative content but also states several action recommendations directly or indirectly. Some of these can be found in the result section and are specific (when to stop procedures, how to position the patient, giving supplements).

The advice to provide sport drinks has not been supported by evidence. Frequently consuming sport drinks, which are high in carbohydrates, can influence birth weight[1]. On the other hand, Madjunkova et al. recommend sport drinks to increase fluid intake, prevent dehydration and add electrolytes[2]. During labor isotonic sport drinks reduce maternal ketosis[3]. I propose to add evidence for sport drinks in vomiting patients or remove advice. It might be better to focus on the main topic of local anesthetics in pregnancy.

Please revise the statement in line 198 "especially between the third and eighth trimesters of pregnancy" as "eighth trimesters" is an typographic error.

1. Grundt, J. H., Eide, G. E., Brantsaeter, A. L., Haugen, M., & Markestad, T. (2017). Is consumption of sugar-sweetened soft drinks during pregnancy associated with birth weight?. Maternal & child nutrition, 13(4), e12405. https://doi.org/10.1111/mcn.12405

2. Madjunkova, S., Maltepe, C., & Koren, G. (2013). The Leading Concerns of American Women with Nausea and Vomiting of Pregnancy Calling Motherisk NVP Helpline. Obstetrics and gynecology international, 2013, 752980. https://doi.org/10.1155/2013/752980

3. Kubli, M., Scrutton, M. J., Seed, P. T., & O'Sullivan, G. (2002). An evaluation of isotonic "sport drinks" during labor. Anesthesia and analgesia, 94(2), . https://doi.org/10.1097/00000539-200202000-00033

Additional comments

The field has been reviewed in the past[4,5]. One recent review article is in press and not yet published[6]. One strength of this review is that in addition to looking at the physiological basics, it offers specific recommendations for the use of local anesthetics during pregnancy (including preparation and injection techniques). This distinguishes the work and sets it apart from other reviews.

4. Lee, J. M., & Shin, T. J. (2017). Use of local anesthetics for dental treatment during pregnancy; safety for parturient. Journal of dental anesthesia and pain medicine, 17(2), 81–90. https://doi.org/10.17245/jdapm.2017.17.2.81

5. Fayans, E. P., Stuart, H. R., Carsten, D., Ly, Q., & Kim, H. (2010). Local anesthetic use in the pregnant and postpartum patient. Dental clinics of North America, 54(4), 697–713. https://doi.org/10.1016/j.cden.2010.06.010

6. Maria A. Manautou, Melanie E. Mayberry (2023). Local Anesthetics and Pregnancy. A review of the evidence and why dentists should feel safe to treat pregnant people. Journal of Evidence-Based Dental Practice, In Press, Journal Pre-proof. https://doi.org/10.1016/j.jebdp.2023.101833.

·

Basic reporting

The text is written in clear and professional English. The article is easy to read and has sufficient literature review. The figures presented are informative and illustrate the discussed topic.
The field of local anesthesia in pregnant women has not been reviewed recently and the publication will be useful for the general dentists.

Experimental design

no comment

Validity of the findings

no comment

Additional comments

To revise - on page 10, I would advise to check the text for mistake in the sentence "Approximately 60% to 75% of pregnant women have gingivitis, especially between third and eighth trimesters of pregnancy(60).
Third and eighth trimester - to be checked for mistakes. There is no eighth trimester of pregnancy.

Reviewer 3 ·

Basic reporting

The theme proposed in the article is interesting, although it is not proposed to evaluate the quality of evidence, it synthesizes knowledge about local anesthesia in Dentistry. However, it is important to use current references, as recommendations have changed over time. Currently, protocols from various institutions have recommended dental care in all trimesters, including prevention, treatment and health education actions (such as ADA). Current publications need to encourage this change in thinking.

GENERAL: I suggest that the review be based on the most current recommendations, such as on the most opportune period of care, recommended procedures, among others. It is necessary to differentiate the use of local anesthetics in dentistry and medicine, because the doses and concentrations are very different. The initial topics give the impression that anesthesia is not safe in pregnancy, contradicting itself with the following topics.

INTRODUCTION
Line 56-58. Currently it is not recommended to postpone dental treatment during pregnancy (educational, preventive and restorative actions). Any improvement in women's oral health and health education can impact children's health.
- The introduction should bring the importance of the proposed review, mainly to clarify doubts of health professionals, because it is a compiled with accessible language. Therefore dental care still causes insecurity among dentists (see: https://www.karger.com/Article/Abstract/481407).

Experimental design

- It is insufficient to allow it to be replicated. It does not provide details of period accomplished, keywords used, or type of studies included. ( does not include any of the criteria evaluated in the topic "Study design").

Validity of the findings

- There are some concepts that need to be better explained, for example, the recommendation that considers the second trimester the most opportune for dental care is based on the comfort of the pregnant woman and not on the risk of treatment for the mother and fetus. Use current references, for example, reference 28 is from 2007.

- The topic "Physiology of pregnancy and anesthesia considerations" seems to me to be more appropriate for a general article on oral health and dental care during pregnancy. As the focus is on anesthesia, it seems more appropriate to focus on directly related topics.

Topic Teratogenicity - it is important to note that not all references used addressed the use of local anesthetics in pregnancy or in humans
- Was the toxicity of local anesthetics considering the correct dose and technique? It would be interesting to address this issue.
- Line 245-246: Which reference supports the last sentence? Are these risks associated with dental anesthesia? With the right technique? Review the text so that it does not generate these doubts/insures for the reader.
- Lines 249-250. phrase "which can affect the development of the fetus's cardiovascular and nervous systems." is unreferenced.
-Topic "Hypoxia and asphyxia": Caution should be written, because the articles used do not refer to doses and concentrations used in dentistry, but rather in the medical area. Flag this to the reader.
Topic "The timing of the operation"
Line 290-292. Reference 28 is 2007. It is important to use a more current recommendation.
- Line 294 - 295. The reference is 2004. It is important to use a more current recommendation.
- 297-303.What is the reference that supports the paragraph? It is important to use a more current recommendation


- I recommend that the authors review the rest of the text and be based on current references and differentiate articles aimed at medical anesthesia

Additional comments

- An interesting topic would show the reader the evolution of this recommendation

---

## Round 0.2 · accepted · Accept

The authors revised the manuscript according to the indications of the reviewers. I consider its acceptance possible.

·

Basic reporting

Overall, the strength of the article lies in the broad view on the topic. An extensive literature base was compiled. The figures and tables support the main content of the article.

Experimental design

The description of the methodology has been adjusted and is now more transparent.

Validity of the findings

This review manages to comprehensively discuss the use of local anesthetics for dental treatment in pregnant patients.

·

Basic reporting

No comment

Experimental design

No comment

Validity of the findings

No comment

Additional comments

The recommended changes have been done in this second edition.